# BCG and SARS-CoV-2—What Have We Learned?

**DOI:** 10.3390/vaccines10101641

**Published:** 2022-09-30

**Authors:** Jakub Kulesza, Ewelina Kulesza, Piotr Koziński, Wojciech Karpik, Marlena Broncel, Marek Fol

**Affiliations:** 1Department of Internal Diseases and Clinical Pharmacology, Medical University of Lodz, Kniaziewicza 1/5, 91-347 Lodz, Poland; 2Department of Rheumatology and Internal Diseases, Medical University of Lodz, Żeromskiego 113, 90-549 Lodz, Poland; 3Tuberculosis and Lung Diseases Outpatient Clinic, Health Facility Unit in Łęczyca, Zachodnia 6, 99-100 Łęczyca, Poland; 4Department of Immunology and Infectious Biology, Faculty of Biology and Environmental Protection, University of Lodz, Banacha 12/16, 90-237 Lodz, Poland

**Keywords:** *Mycobacterium bovis* bacillus Calmette-Guérin (BCG), SARS-CoV-2, COVID-19, immunity, vaccination, non-specific protection, trained immunity

## Abstract

Despite controversy over the protective effect of the BCG (Bacille Calmette-Guérin) vaccine in preventing pulmonary tuberculosis (TB) in adults, it has been used worldwide since 1921. Although the first reports in the 1930s had noted a remarkable decrease in child mortality after BCG immunization, this could not be explained solely by a decrease in mortality from TB. These observations gave rise to the suggestion of nonspecific beneficial effects of BCG vaccination, beyond the desired protection against *M. tuberculosis*. The existence of an innate immunity-training mechanism based on epigenetic changes was demonstrated several years ago. The emergence of the pandemic caused by the severe acute respiratory syndrome coronavirus (SARS-CoV-2) in 2019 revived the debate about whether the BCG vaccine can affect the immune response against the virus or other unrelated pathogens. Due to the mortality of the coronavirus disease (COVID-19), it is important to verify each factor that may have a potential protective value against the severe course of COVID-19, complications, and death. This paper reviews the results of numerous retrospective studies and prospective trials which shed light on the potential of a century-old vaccine to mitigate the pandemic impact of the new virus. It should be noted, however, that although there are numerous studies intending to verify the hypothesis that the BCG vaccine may have a beneficial effect on COVID-19, there is no definitive evidence on the efficacy of the BCG vaccine against SARS-CoV-2.

## 1. Introduction

Although the outbreak of a pandemic was predicted to be an event that was only a matter of time, the emergence of the SARS-CoV-2 virus in 2019, especially the speed at which it reached the pandemic status, came as a big surprise and laid bare the level of unpreparedness for this type of situation [1,2,3]. While the earlier emergence of respiratory viruses—coronavirus causing severe acute respiratory syndrome (SARS-CoV) in 2002 (first reported in the Guangdong province in southern China), which infected 8098 people in nearly 30 countries and killed 774 of them [4,5], and Middle East respiratory syndrome coronavirus (MERS-CoV) in 2012 (first identified in Saudi Arabia), causing 850 cases in over 20 countries [6]—can be read as a harbinger of events to come, this does not alter the fact that determining the precise timing and location of the coronavirus outbreak that led to the current pandemic remains a virtually impossible task. It should come as no surprise, therefore, that the initial phase of the pandemic lacked one of the basic tools to control the situation—a SARS-CoV-2 specific vaccine. In this situation, the existing vaccines appeared to be one of the options that may offer hope of curbing the spread of the coronavirus. The natural choice seemed to be the BCG vaccine, a preparation with a century-long history [7]. Indeed, it had already been shown that BCG vaccination could lead to the development of non-specific protection based on immune training of innate immune cells [8,9,10]. It is therefore not surprising that the investigation of the usefulness of BCG in the protection against SARS-CoV-2 has become a subject of growing interest.

## 2. BCG Vaccine and Its Potential

The BCG vaccine is, as of now, the only approved anti-tuberculosis (TB) vaccine. It was used for the first time on 18 July 1921 to immunize (oral route) a healthy infant born to a woman suffering from TB, who died shortly after the baby’s birth [11]. *Mycobacterium bovis* (bovine mycobacteria), on which the BCG vaccine is based, after 11 years of attenuation (from 1908 to 1919), was deprived of its pathogenicity, which allowed it to be used for the production of an antimycobacterial vaccine [11,12,13,14,15,16]. By 1928, 114,000 children had been vaccinated [11], and although the studies at that time could hardly be assumed to meet the standards of double-blind placebo-controlled trials, the vaccine was considered safe and effective in children [17,18,19,20,21,22]. Soon, vaccination was implemented throughout Europe, and after World War II, the WHO recommended expanding the vaccination campaign beyond Europe [23,24,25].

It is estimated that, as of now, more than 4 billion doses of the BCG vaccine have been administered worldwide [26]. Unfortunately, despite the widespread use of TB vaccination and the announcement of the tuberculosis programs (e.g., DOTS—Directly Observed Therapy Shortcourse, End TB Strategy) by WHO, TB continues to take a heavy toll, with an estimated 10 million new cases and 1.5 million deaths in 2020, which indicates that it is the second-most infectious killer after COVID-19 [27,28,29]. The protective properties of BCG against pulmonary TB vary enormously depending on the region of the world, from 0% in south India [30] to 80% in Norway, Sweden, and Canada [31]. Meta-analyses performed in 1994 showed that the BCG vaccine reduces the risk of tuberculosis by an average of 50%. The protection has been examined in different populations, with different testing methods, and in different types of TB [32]. It has been noted that the BCG vaccine evidently reduces the risk of tuberculous meningitis and disseminated (miliary) tuberculosis, even by 75–87% [33,34,35]. Furthermore, a meta-analysis of experimental and observational studies has indicated, respectively, a 26% (95% CI 14–37%) and 61% (95% CI 51–70%) effectiveness of BCG in the prevention of leprosy [36], and a recent study conducted in Paraguay showed an 89.5% leprosy risk reduction (95% CI, 55.2 to 98.1) [37].

When considering the efficacy of any vaccine, it is very difficult to determine whether its protective effect is the result of preventing infection with an infectious agent or rather preventing the progression from infection to the development of a clinically manifested disease. Hence, there is a need for a thorough understanding of the immunological processes that accompany vaccination which would enable the development of vaccines against hard-to-target pathogens, both those already known (e.g., *M. tuberculosis*) and those newly emerging (e.g., SARS-CoV-2) [38]. Soysal et al. [39], for the very first time, demonstrated that the BCG vaccine prevents not only tuberculosis disease development but also the acquisition of TB infection. Using both a T-cell-based enzyme-linked immunospot assay (ELISpot), not confounded by BCG vaccination, and TST (Tuberculin Skin Test) to assess infection amid 979 children (up to 16 years old) exposed to *M. tuberculosis* (household contact), of whom 79% had a BCG scar, it was found that 110 of 209 (53%) unvaccinated children were deemed to be infected on the basis of ELISpot results, whereas in the group of 770 vaccinated children, the infection was diagnosed in 306 individuals (40%) [39].

Just eleven years after the first use of the BCG vaccine, there appeared the first publication reporting that a marked decrease in mortality among vaccinated children compared to those who were not immunized cannot be explained only by a protective effect of the vaccine, suggesting that “one could evidently be tempted to find an explanation for this lower mortality among vaccinated children in the idea that BCG provokes a non-specific immunity” [40,41,42]. A series of controlled trials from 1948 to 1961 conducted in the US and UK on different age groups (0–21 years) showed a decrease in mortality from diseases other than tuberculosis by an average of 25% (95% CI 6% to 41%) [43]. One of the recent studies of the nonspecific effect of BCG vaccination was conducted in Guinea-Bissau, where the randomized trial between 2004 and 2008 included 2320 low-birth-weight children. A significant decrease (25%) in mortality was shown for children weighing less than 1.5 kg, with lower DTP (Diphtheria-Tetanus-Pertussis) vaccination coverage and fewer cases of neonatal sepsis, respiratory infection, and fever. Thus, BCG vaccination may lead to beneficial effects, especially in the neonatal period [44]. Interesting data were provided by the observational study with a randomized BCG trial conducted between 2015 and 2017 at the Neonatal Intensive Care Unit in Guinea-Bissau and published this year by Schaltz-Buchholzer et al. [45]. It was reported that there is an association between maternal BCG vaccination (the presence of a BCG scar) and a decreased neonatal mortality risk (1.7% for neonates born to mothers with a scar vs. 3.3% for those born to mothers with no scar) [45]. Arts et al. [46] described the increased expression of activation-related surface markers and the enhanced production of cytokines, such as IL-1β, IL-6, IFN-γ, and TNF, by monocytes from adults with a history of BCG vaccination in response to infection with various pathogens compared with monocytes from adults who have not received the BCG vaccine [46]. As a general principle, vaccines are based on the phenomenon of immunological memory associated with adaptive immunity and manifested primarily by the production of antibodies highly specific to vaccine antigens, including the most desirable neutralizing antibodies. Although, for some pathogens, this strategy provides durable immunity (e.g., against polio, measles, or, earlier, against smallpox), in the case of viruses undergoing intensive mutations, it may fail (e.g., influenza virus, or the recently emerged pandemic SARS-CoV-2 virus), entailing the development of more and more “updated” versions of vaccines [47].

How does the BCG vaccine work then (Figure 1)? It is thought that BCG mediates immunity through the development of antigen-specific memory T cells, which act rapidly after subsequent infection with tubercle bacilli [48]. Once the bacteria are injected intradermally, they are recognized by macrophages and dendritic cells (DC) via the interaction of pattern recognition receptors (PRR), e.g., complement receptors (CR) 3 and 4, toll-like receptors (TLR) 2 and 4, as well as other types of receptors: nucleotide-binding oligomerization domain (NOD)-like receptors (NOD2) and intercellular adhesion molecule-3-grabbing nonintegrin (DC-SIGN) with pathogen-associated molecular patterns (PAMPs), e.g., peptidoglycan, arabinogalactan, and mycolic acids, which are present in the bacterial cell envelope [9,48]. The presentation of mycobacterial antigens and the priming of T lymphocytes take place in the nearest lymph nodes to which DCs migrate. Both CD4^+^ and CD8^+^ T cells are activated, and they undertake an increased production of interferon (IFN)-γ, as well as TNF-α and IL-2. IFN-γ enhances the antituberculous activity of macrophages and participates in the activation of B cells, producers of specific antibodies. Granzymes and perforins produced and secreted by CD8^+^ T cells determine the cytotoxic activity of these cells. It is indicated that both BCG-specific CD4^+^ and CD8^+^ T cells are essential for the effective control of mycobacterial infection. Within these two lymphocyte populations, some cells are transformed into memory cells capable of generating a robust lymphoproliferative response to mycobacterial antigens and the intensive secretion of IFN-γ [9]. The role of antibodies induced by the BCG vaccine cannot be completely overlooked either, because, although their importance in the protection against TB is still debatable, their ability to opsonize mycobacterial cells may significantly increase the efficiency of bacterial uptake by phagocytes, limiting the development of infection [49].

First proposed in 2011 by Natea et al. [57], the concept of “trained immunity” describes a long-term functional reprogramming of innate immune cells that is triggered by exogenous or endogenous stimuli and leads to an altered response to a second challenge. The secondary response to a subsequent nonspecific stimulus (related or nonrelated to the first one) may be altered, and, in consequence, the cell response can be more effective or less effective than the primary response [50,58,59,60]. The phenomenon of the training of innate immune cells has broken the pattern of thinking about this type of immunity as being non-enhanced. Immune training develops a type of immune memory of the innate immune system and, through changes in cellular metabolism (e.g., the increase in glycolysis [61], glutaminolysis [62], and cholesterol synthesis [63]) and through epigenetic reprogramming (e.g., histone methylation: H3K4me1, H3K4me3, and acetylation H3K27Ac), which is responsible for chromatin relaxation and increased gene transcription [64], leads to altered cell activity. In the case of BCG, particular attention is paid to the role of NOD2. After being engulfed by a phagocytic cell, BCG undergoes defragmentation in the autophagolysosome, and the released muramyl dipeptide interacts with NOD2, which generates the epigenetic reprogramming [50]. Through NOD2 signaling, epigenetic modifications occur, manifested by an increase in histone H3K4 trimethylation and a decrease in H3K9 trimethylation, which has an impact on the production of proinflammatory cytokines such as TNF-α, IL-1β, and IL-6 [51]. Furthermore, it has been shown that BCG can induce type I IFN production in both mouse and human primary macrophage cells, although at a lower level than tubercle bacilli do. It is hypothesized that BCG vaccination may very subtly tune the production of IFN and other pro-inflammatory cytokines, preventing the undesirable and often devastating overproduction of these in the course of SARS-CoV-2 infection, as detailed by Wannigama and Jacquet [51]. BCG-induced immune training is of particular importance in light of studies showing that epigenetic reprogramming is possible at the stage of hematopoietic stem cells (HSC) and multipotent progenitors (MPP). It was demonstrated on a mouse model that, after the intravenous administration of BCG, the bacteria reached the bone marrow and caused an increase in the number of LKS^+^ (c-Kit^+^ Sca-1^+^) progenitor cells and the generation of epigenetically modified macrophages derived from “BCG-educated” HSC, with increased protective potential against *M. tuberculosis*. BCG-triggered local cell expansion and enhanced myelopoiesis, at the expense of lymphopoiesis, ultimately result in the appearance of long-term innate immune protection associated with the transcriptional reprogramming of gene expression by the epigenetic machinery. It is suggested that, through HSC targeting, the development of new-generation vaccines will be possible [65].

## 3. Coronaviruses and SARS-CoV-2

The coronavirus family (Coronaviridae) is the largest of the order Nidovirales and consists of two subfamilies: Letovirinae and Orthocoronavirinae. There are four genera within the Orthocoronavirinae subfamily: alpha-, beta-, gamma-, and deltacoronaviruses [66,67,68]. Alpha- and betacoronaviruses are found in mammals, while gamma- and deltacoronaviruses occur primarily in birds. Coronaviruses have positive-sense, single-stranded genomes made of RNA, with a spherical structure and an 80–120 nm diameter. Compared to other RNA viruses, coronaviruses have the largest 30-kilobase genome [69]. In animals, they cause respiratory, gastrointestinal, hepatic, and nervous system diseases [70] (e.g., infectious peritonitis in cats, epidemic diarrhea in pigs, viral gastroenteritis in cattle, or infectious bronchitis in fowl). Historically, coronaviruses were thought to cause mild cold-like illnesses in humans. However, in 2002, a coronavirus named after severe respiratory syndrome, SARS-CoV, emerged in China. The SARS-CoV epidemic lasted 8 months and resulted in 8098 confirmed human cases worldwide, of which 774 (9.5%) were fatal. About 10 years after the emergence of SARS-CoV, another highly pathogenic human coronavirus, MERS-CoV, emerged in Saudi Arabia. To date, there are four coronaviruses that have been found to cause mild respiratory symptoms in humans—HCoV-OC43, HCoV-NL63, HCoV-HKU1, and HCoV-229E—and two coronaviruses—SARS-CoV and MERS-CoV—thatinduce acute respiratory infections [66,67]. Human coronaviruses (HCoV) are classified as α-CoV (HCoV-229E and NL63) and β-CoV (MERS-CoV, SARS-CoV, HCoVOC43, and HCoV- HKU1) [71]. The emergence of new, highly pathogenic coronavirus species and the understanding that these viruses in children, the elderly, and people with immunodeficiencies can lead to the development of serious, life-threatening diseases [72] have evoked an increased interest in these pathogens.

In December 2019, a new coronavirus, SARS-CoV-2, was identified in Wuhan, China and become responsible for a pandemic of respiratory infections worldwide. SARS-CoV-2 is a β-CoV, and although genetic studies have shown it to be closely related to two bat CoV-like coronaviruses (bat-SL-CoVZC45 and bat-SL-CoVZXC21), its genome is essentially similar to that of typical CoVs [73].

The origin of SARS-CoV-2 is not entirely clear, but recent debate has focused around two competing hypotheses: a “laboratory escape” and a zoonotic emergence [74]. In contrast to epidemiological links to animal markets in Wuhan, so far, there is no sufficiently convincing evidence that any early cases of infection could be linked to the Wuhan Institute of Virology (WIV). There is also no evidence that WIV possessed or was working on SARS-CoV-2 prior to the pandemic outbreak. The suspicion that SARS-CoV-2 may have a laboratory origin stems rather from the coincidence that the virus was first detected in a city with a large virology laboratory studying coronaviruses [74]. However, the ambiguous attitude of the Chinese authorities—manifested by limiting information on the course of the epidemic, especially in its earliest period, and blocking access to documents on the course of the SARS epidemic in 2001–2004—isnot conducive to ruling out the lab leak scenario. Although Kristian Andersen argues in his study that there are no traces in the virus genome indicating genetic manipulation [75], other virologists point to some clues, such as the presence of the furin cleavage sites, which the closest relatives of SARS-CoV-2 do not have, or a combination of nucleotides that underlie a segment of the furin cleavage site: CGG (encoding arginine), which, according to David Baltimore, a Nobel laureate and professor emeritus at the California Institute of Technology in Pasadena, is rather uncommon in viruses but is often a derivative of intentional genetic manipulations [76]. Holmes et al. [74] argue that, although an animal reservoir for SARS-CoV-2 has not been identified and key species may not have been studied, unlike in other scenarios, there is nonetheless ample scientific evidence supporting a zoonotic origin. While the possibility of a laboratory “accident” cannot be completely eliminated, it seems the least likely when compared to the numerous and repeated human–animal contacts that occur routinely in the wildlife market [74]. The existence of an intermediate host from which the microorganism can be transmitted to humans has been demonstrated for SARS-CoV—civets—and for MERS-CoV—dromedaries. Human-to-human transmission occurs primarily through the aerogenic route, direct contact with an infected person, or through contact with contaminated objects and surfaces [77]. Surprisingly, SARS-CoV-2 can effectively infect both the upper and lower respiratory tracts. In contrast to SARS-CoV, SARS-CoV-2 has been observed to replicate in neurons, leading to significant neurological symptoms such as confusion and disturbances/loss of smell and/or taste, which are rarely reported in patients with SARS-CoV [78].

Four major structural proteins of SARS-CoV-2 needed for the invasion of host cells have been identified: spike/surface glycoprotein (S), envelope protein (E), which is responsible for virus formation, membrane glycoprotein (M), which determines the shape of the viral envelope, and nucleocapsid protein (N), having both a protective function for RNA genetic material and an active role in modifying cellular processes and viral replication. Glycoprotein S (spike) is a homotrimer anchored in a viral envelope and forms elements that protrude from the surface of the virus. They facilitate the binding of the virus to host cells via angiotensin-converting enzyme 2 (ACE2), which is expressed in the airway epithelia [77,79,80,81].

According to data from WHO, as of 20 September 2022, there have been 609,848,852 confirmed infections worldwide, including 6,507,002 deaths [82]. The immune response to SARS-CoV-2 virus invasion involves both cellular and humoral—as well as innate and acquired—mechanisms (induction and maturation of dendritic cells, increased production of pro-inflammatory cytokines and mediators by CD4^+^ T cells, production of virus-specific IgM and IgG immunoglobulin by B cells, direct destruction of infected host cells by CD8^+^ T cells, production of complement factors), as detailed by Mortaz et al. [83] and Bhardwaj et al. [84]. Unfortunately, for reasons that are not fully understood, some infected individuals develop an excessive/anomalous immune response, leading to complications with very serious consequences for the patient [85,86].

History teaches that widespread immunization is the best solution to reduce/stop the epidemics. This was the case, for instance, with smallpox andTB. Vaccines against COVID-19 are now also available. They are based on different concepts: mRNA vaccines (Comirnaty by Pfizer-BioNTech, Spikevax by Moderna), vaccines using an adenoviral vector (Vaxzevria by AstraZeneca, Vaccine Jannsen by Johnson&Johnson), and “classic” protein vaccines (Nuvaxovid by Novavax) [87]. The latest advances in COVID-19 vaccine research are discussed in [88,89,90]. Before coming into the market, the BCG vaccine began to be the object of great interest because of the beneficial effect of “trained immunity” it generated. The emergence of the SARS-CoV-2 virus pandemic initiated a debate on whether the BCG vaccine could influence the immune response against the virus. Research in this area, eagerly funded before the advent of vaccines that elicit a specific protective response, is now more often seen as less promising—wrongly so, because, given the mortality rate for COVID-19, it is important to properly investigate every factor that may have potential protective value against severe infection, complications, and death.

## 4. Does BCG Have the Potential to Reduce the Pandemic’s Impact?

Before specific vaccines became available, a 100-year-oldBCG vaccine had been proposed as a hypothetical method for preventing COVID-19—a new coronavirus disease in humans [91]. Through its ability of strong and long-lasting activation of the immune system, the BCG vaccine is able to enhance the defense mechanisms to effectively cope with infections other than BCG. This is due to the antigen-independent activation of B and T cells—this mechanism is referred to as heterologous immunity [92], and the reprogramming of innate immune cells is called trained immunity [93].

Using the virtual tool Protein BLAST (basic local alignment search tool), several similar 9-amino acid sequences between *Mycobacterium bovis* BCG and SARS-CoV-2 were identified [52]. The differences involve only a single amino acid within the entire sequence, e.g., VLGGLAATV in BCG and VLGSLAATV in SARS-CoV-2 (Figure 1). Analysis with the use of two computer algorithms, NetMHCpan 4.1 (which allows for the evaluation of whether the peptide can bind to HLA class I and be recognized by cytotoxic T lymphocytes, CTL) and IEDB (which allows for the prediction of the T cell epitops), revealed that some of the 9-amino acid sequences have good binding affinity for common HLA class I molecules. Since CTLs (CD8^+^ effector T cells) play a key role in destroying virus-infected host cells, and considering that BCG has some 9-amino acid peptide sequences showing great similarity with those identified in SARS-CoV-2, it can be hypothesized that the BCG vaccine has the potential to generate cross-reactive T cells against SARS-CoV-2 [52].

Extremely interesting results were provided by Counoupas et al. [94], who proposed an experimental BCG:CoVac vaccine—a combination of BCG with a stabilized, trimeric form of the SARS-CoV-2 spike (Spk) antigen. It was shown in an animal model (K18-hACE2 transgenic mice expressing human ACE2) that the administration of BCG:CoVac provoked the appearance of virus-specific IgG antibodies in the blood of vaccinated mice. Notably, the induction of neutralizing antibodies with high affinity to SARS-CoV-2 and the release of Th1-type cytokines were noted. The appearance of these humoral immune response elements corresponded with the appearance of Th cells in local lymph nodes and with increased levels of antigen-specific plasma B cells after vaccination. The administration of as little as a single dose of BCG:CoVac led to an almost complete abolition of disease symptoms after the SARS-CoV-2 challenge (minimal inflammation and no detectable level of the virus in the lungs of infected animals). Furthermore, the immunization effect was extended and enhanced when mice previously immunized with BCG:CoVac were boosted with a heterologous vaccine (SpK formulated in Alhydrogel/alum (AlmSpK)). This led to an enhanced production of SARS-CoV-2-specific antibodies that effectively neutralized the SARS-CoV-2 variants of particular concern: B.1.1.7 and B.1.351. These findings demonstrate that BCG-based vaccination has the potential to protect against the major SARS-CoV-2 variants circulating worldwide [94].

### 4.1. Cross-Protective Immunity Brings Hope

The cross-protective effect of the BCG vaccine in non-TB diseases has been demonstrated in several studies. It has been shown that BCG vaccination improves the response to the human papillomavirus, herpes simplex virus [10,95], and even to malaria [96]. An immune training mechanism is responsible for this effect, at least in part, in which the epigenetic reprogramming of innate immune cells leads to the increased production of inflammatory cytokines and, consequently, a stronger immune response [15]. Kurthkoti et al. [22] hypothesized that BCG vaccination provides heterologous immunity by enhancing the innate and adaptive immune response to COVID-19. They concluded that it is possible that prior BCG vaccination may result in higher levels of IFN-γ, which may prevent T-cell migration to the lungs and reduce tissue damage [97].

Interestingly, it is postulated that not only the BCG vaccine but also other vaccinations such as influenza and even pneumococcal vaccination can be helpful in the fight against COVID-19 [98]. A meta-analysis of 23 trials with a total of 1,037,445 participants showed that influenza vaccination was associated with a reduced risk of COVID-19 infection (RR = 0.83, 95% CI = 0.76, 0.90) and hospitalization (RR = 0.71, 95% CI = 0.59, 0.84), and it was not associated with intensive care unit admission or death (risk of ICU admission: RR = 0.93, 95% CI = 0.64, 1.36; risk of death: RR = 0.83, 95% CI = 0.68, 1.01). Furthermore, it was reported that the use of the quadrivalent influenza vaccine may be associated with a reduced risk of COVID-19 infection (RR = 0.74, 95% CI = 0.65, 0.84) [99]. This suggests that non-COVID-19-specific vaccination may be useful against COVID-19.

There are many studies analyzing the COVID-19 cases and deaths as well as the pandemic dynamics in countries that widely vaccinate their populations with BCG compared to non-vaccinating countries. Pavan Kumar et al. [100] showed that BCG vaccination in a group of healthy elderly people aged 60–80 years resulted in a reduction in the plasma levels of pro-inflammatory cytokines and chemokines, which are also increased in severe SARS-CoV-2 infections. The study demonstrated the immunomodulatory properties of BCG vaccination, which may play a protective role against inflammatory diseases in the elderly population [100]. Sarfraz et al. [101] (Table 1) analyzed mortality and death dynamics due to COVID-19 during the initial phase of the pandemic in the context of the presence or absence/limitation of a BCG vaccination program. A specially designed methodology was used to calculate mortality rates, taking into account two reference points: the date on which at least 100 confirmed cases were reported in a country (with 31 March as the cut-off date) and the 30-day period from the 100th case-day. These data were related to the area of the country and its population. It turned out that the first 10 countries with the highest death rate recommend BCG only for certain narrow groups of patients in their national vaccination programs. In contrast, the last 10 countries in the ranking (with the lowest dynamics of deaths in the initial phase) offered BCG vaccination for all citizens at birth [101]. In addition, in a paper by Charoenlap et al. [102], it was reported (based on epidemiological data) that countries with a current universal BCG policy have a lower median mortality compared to countries that have had a universal BCG policy in the past [102]. Amirlak et al. [103] described a study conducted in early March 2020 in the United Arab Emirates, in which a booster dose of the BCG vaccine was administered to healthcare personnel. The COVID-19 infection rate in the unvaccinated group was 8.6% compared with 0% in the vaccinated group, demonstrating the potential effectiveness of the BCG booster in a high-risk healthcare population [103]. Furthermore, the study by Rivas et al. [104] on a cohort of 6201 healthcare workers at a Los Angeles hospital (of whom 29.6% had been vaccinated against BCG and 68.9% had not) showed there was a significant correlation between the BCG vaccination of healthcare workers and a lower morbidity, as well as lower self-reported symptoms among patients [104].

Senoo et al. [112], reviewing the morbidity and mortality rates at different stages of the pandemic and the differential impact of BCG implementation in the countries of the Organization for Economic Cooperation and Development (OECD), also pointed to the protective value of BGC in the context of COVID-19 [112]. Interesting data have been provided by a study published in 2021 by Pathak et al. [105] (Table 1), who analyzed COVID-19 mortality rates through a broader prism—vaccination programs not only against TB but also against influenza. The evaluation considered three time points during the initial phase of the pandemic (17 May, 1 October, and 31 December 2020) in countries that reported at least 1000 deaths caused by SARS-CoV-2. The countries with high influenza vaccination rates observed high mortality rates, and the countries vaccinating against TB but not against influenza noted lower numbers of COVID-19 deaths/million compared to the countries vaccinating against influenza but not TB. It was found that immunization with BCG negatively correlates with COVID-19 deaths/million [105]. The lower incidence of COVID-19 in the countries with routine BCG vaccination programs (most likely a consequence of non-specific vaccine effects) is also highlighted by Mohapatra et al. and Pandita et al. [113,114]. An intriguing conclusion that every 10% increase in the BCG index is associated with a 10.4% reduction in COVID-19 mortality was formulated by Escobar et al. [115] regarding the potential relationship between BCG vaccination and COVID-19 after analyzing epidemiological data and taking into account factors such as the stage of the COVID-19 epidemic, development, rurality, population density, and age structure in different countries. The authors of the study claim, however, that although the beneficial effect of BCG is noticeable, the data obtained should be treated with caution due to broad differences between countries [115]. In another study, Kumar et al. [116] focused on BCG vaccination and temperature as predictor variables for both newly diagnosed COVID-19 cases and mortality. However, no statistically significant association was observed between median temperature and virus transmission. Nevertheless, the rate of spread of SARS-CoV-2 infection was significantly lower in countries with BCG vaccination policies [116]. In a Letter to the Editor published in 2020, Ozdemir et al. [117] demonstrated a statistically significant difference in terms of COVID-19 cases and mortality between the populations of countries with consistent BCG vaccination policies (137) and the populations of countries with limited or never-implemented vaccination programs (37), in favor of the former (0.0147 ± 0.027 vs. 0.1892 ± 0.244, *p* < 0.0001;0.0004 ± 0.001 vs. 0.0113 ± 0.020, *p* < 0.0001, respectively) [117]. This is in line with the observations of Miller et al. [118] and Hensel et al. [119], who reported that countries without universal BCG vaccination in their health policy, such as Italy and the USA, experienced higher COVID-19-related mortality than countries with long-standing universal BCG vaccination policies, such as South Korea and Japan [118,119]. Similar conclusions were reached by Gursel and Gursel [120], who compared the number of deaths due to COVID-19 in countries with a national BCG vaccination program and those that discontinued their vaccination procedures. They found that the number of deaths relative to the population size in countries with a national BCG vaccination program was significantly lower than that in those without vaccination [120]. This may suggest that the heterologous non-specific protective effect induced by BCG vaccination may be long-lasting and therefore potentially affect the dynamics of SARS-CoV-2 spread and/or disease severity [120]. The results of multivariate regression tests with predictors such as economic (economic development indicators), health (prevalence of chronic diseases such as type 2 diabetes and cardiovascular mortality), and quantitative (school closures and education management) characteristics related to pandemic containment in countries with existing or previous BCG vaccination policies and countries that provided vaccination only to specific risk groups confirmed the role of BCG in COVID-19 infection. BCG vaccination coverage, especially among recently vaccinated populations, seems to contribute to mitigating the spread and severity of COVID-19 [121,122]. The study by Singh et al. [123] meta-analyzed the mortality rates and cure rates in countries with and without BCG vaccination policies. They found lower COVID-19 mortality rates in countries with an anti-TB vaccination policy compared to countries without one: 1.31% (95%CI—1.31% to 1.32%; I2 = 100%, *p* < 0.01) and 3.25% (95%CI—3.23% to 3. 26%; I2 = 100%, *p* < 0.01), respectively. It was concluded that 52 people need to be vaccinated with BCG to prevent one death by SARS-CoV-2 (NNT = 52). The study supports the hypothesis that BCG vaccination can provide protection against COVID-19 [123].

In addition to many publications based solely on epidemiological data published in the early stages of the pandemic [124,125,126], randomized trials have also been undertaken. On 20 April 2020, a clinical trial entitled BCG Vaccine for Health Care Workers as Defense Against COVID 19 (BADAS) was launched in the United States, with 1800 participants aged 18–75 years. It was hypothesized that BCG vaccination might reduce the risk of SARS-CoV-2 infection among healthcare workers, along with the severity of disease. Hence, some participants were immunized with the BCG vaccine (BCG Tice strain) and some took a placebo. Participants and investigators were blinded. The study was scheduled for completion in May 2022 [127,128].

### 4.2. An Illusory Hope: BCG Does Not Change the COVID-19 Status

Contrary to the above, there are papers that question the thesis of a protective effect of BCG in SARS-CoV-2 infection [129]. The main argument for the lack of an association between BCG vaccination and COVID-19 incidence is that the country where the pandemic began—China—uses universal BCG vaccination [130]. Additionally, other countries hit hard by the new coronavirus such as Iran and Egypt have BCG vaccination rates of 99% and 95%, respectively [129]. Similar conclusions were drawn by Hamiel et al. [106] (Table 1) from a cohort study involving 5933 adults aged 35 to 41 years (3064 vaccinated and 2869 unvaccinated). No statistically significant difference in the rate of COVID-19 positivity between the BCG-vaccinated group and the unvaccinated group was shown [106]. Data from Finland and Australia also appear to contradict the hypothesis that BCG immunization may reduce COVID-19 morbidity. These countries ceased their universal BCG vaccination programs in the mid-1980s (Australia) and 2006 (Finland); yet, they show low COVID-19 mortality/million rates compared to countries with current mandatory BCG vaccination. Thus, BCG vaccination—if it contributes to lower COVID-19-related mortality—is most likely not the only factor. Both countries have excellent healthcare systems and low population densities. The latter factor in particular, along with its derivative, social distancing measures, seem to be far more effective in reducing COVID-19 infections compared to countries with a high population density [131]. The effectiveness of BCG vaccination in reducing morbidity and mortality caused by COVID-19 was investigated in two randomized trials involving healthcare workers in South Africa [107] and Poland [108] (Table 1). The results of these multicenter, randomized, double-blind studies published in 2022 indicated that BCG vaccination was not effective in protecting healthcare workers from SARS-CoV-2 infection or severe COVID-19 and hospitalization. Several other studies also indicate that there is no correlation between BCG vaccination, including that received in childhood [106], and the spread and mortality of SARS-CoV-2 [132,133,134,135,136]. An extremely interesting approach to verifying the hypothesis, formulated on the basis of epidemiological studies, that BCG vaccination may be responsible for the lower morbidity and mortality from COVID-19 recorded in less developed countries was demonstrated by Bates et al. [109]. They analyzed the anonymized records of U.S. Military Veterans treated by the Department of Veterans Affairs, of which 263,039 were classified as not having had COVID-19 (control), while 167,664 records were COVID-19 cases, of which 5016 were deaths. The combination of country and year of birth was used as an indicator of BCG vaccination in infancy. BCG immunization in infancy was shown to have no protective effect against COVID-19. The odds ratio for infection was 1.07 (95% confidence interval (CI): 1.03, 1.11), and the risk ratio for mortality among the COVID-19 cases was 0.86 (95% CI: 0.63, 1.18). Thus, the authors demonstrated that epidemiological studies, while useful for formulating hypotheses, must be verified using other tools such as individual-person measures of exposure and health outcome and randomized trials.

A number of studies are currently underway that will certainly be helpful in exploring this issue further [137]. One of them is a study conducted by Madsen et al. [110] in Denmark, involving 1500 healthcare workers in Danish hospitals, which aims to investigate whether BCG vaccination can reduce susceptibility to SARS-CoV-2 infection and thus prevent absenteeism from work and reduce hospital admissions among participants during the COVID-19 pandemic [110]. A study conducted in Brazil, on the other hand, aims to verify the efficacy and safety of the BCG vaccine in preventing or reducing the incidence of COVID-19 in the city of Goiânia among previously BCG-vaccinated healthcare workers. This is a phase II study on the repositioning of BCG as a preventive strategy against COVID-19 [138].

## 5. Conclusions

A significant number of studies aiming to verify the hypothesis that BCG vaccination may have a beneficial effect on the course of COVID-19, or may even prevent the disease, are observational in nature. They analyze the relationship between the co-occurrence of two factors but do not confirm a causal relationship. Their conclusions may be burdened with error due to the influence of other factors, including the impact of pandemic-related preventive measures (e.g., movement restrictions, social isolation) and the difference in COVID-19 case reporting. In order to find out if the BCG vaccine offers any protection against COVID-19, randomized trials are needed, such as the world’s largest Australian-led BRACE study (BCG vaccination to reduce the impact of COVID-19 in healthcare workers), involving almost 10,000 healthcare workers from the UK, Australia, the Netherlands, Spain, and Brazil. The study was expected to be completed by 31 March 2022 [111]. Similar studies are also underway in France [139], USA [128], Columbia [140], and Egypt [141].

Due to the lack of definitive evidence on the efficacy of BCG against SARS-CoV-2, current WHO guidelines do not recommend BCG vaccination for the prevention of COVID-19 [113], although the beneficial effect of BCG vaccination on the risk and severity of this new disease still cannot be definitively excluded [142]. How the BCG vaccine may confer protective immunity against the pandemic coronavirus must be a subject of further investigation.

## Figures and Tables

**Figure 1 vaccines-10-01641-f001:**
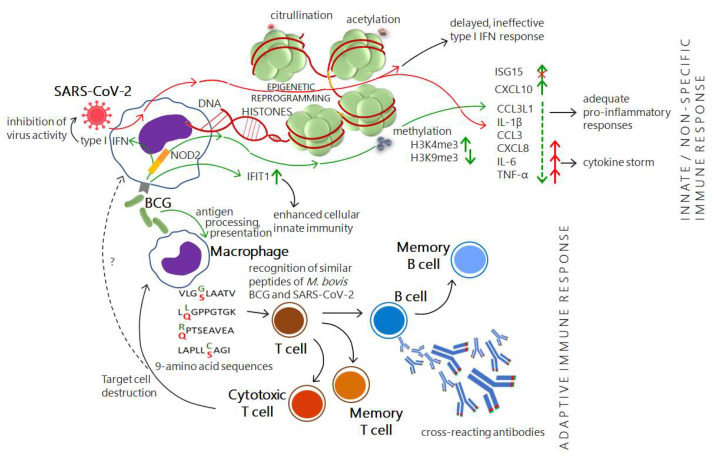
Possible BCG vaccine cross-reaction with SARS-CoV-2 and its mechanism of action. The effectiveness of the immune response consists in the balanced reactivity of both innate and adaptive immunity and their cooperation. It has been shown that cells of innate immunity (for example, macrophages) undergo a process of immune training underpinned by epigenetic changes. BCG, through the induction of epigenetic reprogramming (manifested, for instance, by the production of pro-inflammatory cytokines at appropriate levels), can provide cross-protection during a subsequent infection caused by an infectious agent other than the primary one. Genes, particularly interferon-related, are also subject to epigenetic control by the SARS-CoV-2 virus, the infection of which may be accompanied by the increased production of pro-inflammatory cytokines leading to a cytokine storm. The interferon pathway is one of the most important signaling pathways in the course of infection. In viral infection, an increase in the levels of ISGs is observed, whose stimulation results in the inhibition of viral replication. The BCG vaccine, in particular, leads to the upregulation of ISG15, which may confirm its contribution to the inhibition of viral infection. However, in the case of SARS-CoV-2, its ability to alleviate some ISG15 anti-viral activities was reported. Among the components of the interferon pathway are IFIT proteins. The antiviral properties of IFIT1 are manifested by the recognition of RNA having 5′ triphosphate and lacking 2′-O methylation, followed by the sequestration of the viral nucleic acid. The elevated IFIT1 levels observed after BCG vaccination may indicate its usefulness in mitigating the SARS-CoV-2 infection. Furthermore, it was reported that *M. bovis* BCG contains similar T cells epitops asSARS-CoV-2. The consequence of this can be the production of cross-reactive antibodies by antigen-specific B lymphocytes. Hence, it has been suggested that BCG immunization, as well as latent *M. tuberculosis* infection, may result in an increased effectiveness of the immune system against SARS-CoV-2, not only due to the training of innate immunity but also through the generation of antibodies with cross-reactivity. Abbreviations: CCL3—CC motif chemokine ligand 3 (MIP-1α), CCL3L1—CC motif chemokine ligand 3 like 1, CXCL8/10—CXC motif chemokine ligand 8/10 (IL-8/IP-10), IFIT—interferon-induced proteins with tetratricopeptide repeats, IFN—interferon, IL—interleukin, ISG—interferon-stimulating genes, NOD2—nucleotide-binding oligomerization domain containing 2, TNF-α—tumor necrosis factor alpha. Based on [50,51,52,53,54,55,56].

**Table 1 vaccines-10-01641-t001:** Exemplary studies on the possible connection between BCG vaccination and COVID-19.

Topic/Title	Study Design	Objective /Outcome	References
Variances in BCG protection against COVID-19 mortality: a global assessment.	epidemiological study, an a-priori methodology used to identify the top ten and bottom ten countries according to the death rate (the relative number of COVID-19 deaths within a population per unit of time). For every country until 31 March 2020, which had at least 100 confirmed cases, a tabulated list was arranged with the total deaths within that population from a day at which at least 100 confirmed cases were recorded.	Global assessment of countries according to COVID-19 death rate per population and BCG vaccination policies. COVID-19 mortality may be associated with BCG vaccination as countries like South Korea, Japan, being ones with an active BCG vaccination policy, have a lower COVID-19 mortality rate than countries with no active BCG vaccination policy such as the USA and Italy.	[101]
BCG vaccination history associates with decreased SARS-CoV-2 seroprevalence across a diverse cohort of health care workers	longitudinal, retrospective observational study (healthcare workers, USA)	Multivariate analysis to determine whether a history of BCG vaccination was associated with decreased rates of SARS-CoV-2 infection and seroconversion. A history of BCG vaccination, but not meningococcal, pneumococcal, or influenza vaccination, was associated with decreased SARS-CoV-2 IgG seroconversion.	[104]
Countries with high deaths due to flu and tuberculosis demonstrate lower COVID-19 mortality: roles of vaccinations	epidemiological, multifactor study; Data analysis from the set of 21 (17 May 2020), 51 (1 October 2020) and 83 (31 December 2020) countries regarding TB exposure and BCG policies with respect to COVID-19 incidences and deaths	Countries with high BCG coverage have lower deaths due to COVID-19. COVID-19 deaths are much lower in countries with high flu and TB deaths. Conversely, countries with low flu and TB deaths, in the absence of BCG vaccinations, demonstrate high COVID-19 deaths.	[105]
SARS-CoV-2 rates in BCG-vaccinated and unvaccinated young adults	cohort study, participants born from 1979 to 1981 (national immunization program ended in 1982) and from 1983 to 1985 (Israel)	The comparison of rates of coronavirus PCR test positivity among persons with symptoms suspicious for COVID-19 who did and did not receive BCG vaccination as part of routine childhood immunization in the early 1980s. No evidence to support the hypothesis that BCG vaccination in childhood has a protective effect against COVID-19 in adulthood.	[106]
Safety and efficacy of BCG re-vaccination in relation to COVID-19 morbidity in healthcare workers: A doubleblind, randomised, controlled, phase 3 trial	multicentre, randomised, double-blind, placebo-controlled trial (healthcare workers, South Africa)	BCG does not protect healthcare workers from SARS-CoV-2 infection or related severe COVID-19 disease and hospitalisation.	[107]
Phase III clinical trial evaluating the Impact of BCG re-vaccination on the incidence and severity of SARS-CoV-2 Infections among symptomatic healthcare professionals during the COVID-19 pandemic	multicenter, randomised, double-blind, placebo-controlled (healthcare workers, Poland)	The assessment of re-vaccination against tuberculosis with the BCG-10 vaccine (Biomed Lublin S. A., Lublin, Poland) on an impact on SARS-CoV-2 virus infection and the course of COVID-19 disease (incidence, severity) in healthcare workers with a history of BCG vaccination. No significant correlation between the frequency of incidents suspected of COVID-19 and BCG-10 vaccination, the result of the tuberculin test and the number of scars.	[108]
BCG vaccination at birth and COVID-19: a case-control study among U.S. military Veteran	case-control study of COVID-19 infections with a retrospective cohort study of mortality nested within the infections (anonymized records of U.S. Military Veterans treated by the Department of Veterans Affairs, USA)	No evidence to support the hypothesis that infant BCG vaccination protects against infection or death from COVID-19.	[109]
Using BCG vaccine to enhance non-specific protection of health care workers during the COVID-19 pandemic.	multicentre, placebo-controlled, single-blinded, randomised controlled clinical trial (healthcare workers, Danmark)	The reduction of absenteeism due to illness among healthcare workers during the COVID-19 pandemic. The reduction in the number of healthcare workers that are infected with SARS-CoV-2, and the reduction in the number of hospital admissions among healthcare workers during the COVID-19 pandemic. (in progress)	[110]
The BCG vaccination to Reduce the impact of COVID-19 in Australian healthcare workers following Coronavirus Exposure (BRACE)	multicentre, open label randomized controlled clinical trial (healthcare workers, Australia)	Does BCG vaccine reduce the incidence of symptomatic and severe COVID-19, as well as other respiratory illnesses and allergic diseases? (in progress)	[111]

## Data Availability

Not applicable.

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
