# Peer review of "BCG and SARS-CoV-2—What Have We Learned?"

_vaccines, 2022, doi:10.3390/vaccines10101641_

Round 1
Reviewer 1 Report
Estimated Authors of the paper "BCG and SARS-CoV-2 – what have we learned?"
Thank you for the opportunity to review this interesting paper. In this narrative review, Kulesza have collected and discussed several reports on the potential impact of BCG vaccination in countering the occurrence of new SARS-CoV-2 infections. Following the early suggestions from Ozdemir, they have discussed on the occurrence of SARS-CoV-2 infections and their correlation with BCG vaccination rates through the trained immunity theory.
From the point of view of the present reviewer, the present study cannot be accepted on Vaccines for several reasons, and more precisely:
1) the present study is a narrative review, this substantial shortcoming is not addressed in any section of the paper; this is a particularly significant issue as - across the text, only reports that are consistent with the hypothesis that SARS-CoV-2 infection may be hindered by a previous BCG vaccination are reported and discussed. As no research strategy is preventively reported, the risk of potential "cherry picking" may appear substantial and should be therefore addressed either in the introduction (discussing how the evidences were collected across the available information sources) or in the discussion, through a comparison of the reliability of a narrative review with a systematic review with meta-analysis.
2) in this regard, please be aware that in the early stages of the pandemic, some reports have suggested that some of the alleged preventive effects of BCG vaccine on SARS-CoV-2 infection may have been explained through the diachronous nature of the pandemic (see for example: https://www.mattioli1885journals.com/index.php/actabiomedica/article/view/9700).
3) the text is disproportionately organized, with introductory sections on both BCG and SARS-CoV-2 that are largely unnecessary in this stage of the pandemic. Please shorten them.
Author Response
The authors wish to thank the Reviewer for revising the manuscript and comments.
We would like to clarify that the reviewed manuscript was not intended to prove the thesis of a beneficial effect of BCG vaccine on Covid-19 status. The references included were in no way biased. Furthermore, the authors avoided statements that would definitively identify BCG vaccination as an effective intervention in SARS-CoV-2 infection. Moreover, the authors of the cited publications were themselves very cautious about the results of their work. Hence, in our work we also used terms such as: "...studies have suggested that...". (line 311), "Kurthkoti et al. hypothesized taht...". (line 340), "...properties of BCG, which may play...". (line 360) etc. In line 404 it is explicitly stated that the research results obtained must be treated with great caution. To dispel any doubts, some corrections have been made to the manuscript:
1/ the following sentence was included in the abstract:
“It should be noted, however, that although there are numerous studies to verify the hypothesis that BCG vaccine may have a beneficial effect in COVID-19, there is no definitive evidence on the efficacy of BCG vaccine against SARS-CoV-2.”
2/ in section 4, two subheadings have been added
4.1. Cross-protective immunity brings hope
4.2. An illusory hope…BCG does not change the COVID-19 status
3/ information on symptoms of virus infection was removed from section 3.
The authors faced a daunting task; a radical reworking of the manuscript would in effect result in a de facto new paper, which would stand in stark contrast to the opinions of the other reviewers. It is therefore sincerely hoped that the changes made meet the reviewer's expectation and that the provided clarifications will dispel doubts about the objectivity of the authors of the paper.
Reviewer 2 Report
In this review manuscript, the authors attempted to describe whether the BCG vaccine was useful to mitigate the pandemic impact of SARS-CoV-2. The authors seem to have successfully summarized the contents of many references. Frankly speaking, the major concern of the reviewer has is how many COVID-19 researchers would like to read this review at present when the SARS-CoV-2-specific vaccines such as the mRNA vaccine and Adenovirus-vectored vaccine have been established. Vaccine researchers are currently interested in creating the universal vaccine that is specifically effective against many mutant strains of SARS-CoV-2. However, this review might be significant if the authors propose to consider the BCG vaccine as a temporal bulwark against newly emerging viruses that will cause pandemics in the future until a virus-specific vaccine can be developed.
Specific comments:
1. The authors should rewrite the abstract.
I believe that the authors would like to write there are a significant number of studies aiming to verify the hypothesis that the BCG vaccine may have a beneficial effect in COVID-19, but there is no definitive evidence on the efficacy of the BCG vaccine against SARS-CoV-2. The authors should write this conclusion in the abstract.
2. As described in lines 315-332, it is of interest to use the BCG vaccine as an adjuvant in SARS-CoV-2 specific vaccines. Are there any references showing which component of the BCG vaccine activates innate immunity? Is it possible to use this component as a vaccine adjuvant?
3. The authors should shorten the section “3. Coronaviruses and SARS-CoV-2”. I think detailed information on SARS CoV-2 such as the origin of the virus and symptoms of COVID-19 etc. is not necessary for this review.
Author Response
The authors wish to thank the Reviewer for revising the manuscript and precious comments. The most recent changes in the text have been marked in red.
Following the Reviewer's suggestions:
1/ we have slightly rearranged the Abstract and added the sentence, which was suggested by the Reviewer,
2/ the complete experimental vaccine (BCG:CoVac) referred to in lines 315-332 consisted of BCG cells (5 × 105 CFU), SARS-CoV-2 full-length spike stabilized, trimeric protein (SpK) expressed in EXPI293F™ cells (SpK; 5 μg) and Alyhydrogel (100 μg);
It is very difficult to pinpoint which component of the BCG cell is particularly responsible for the induction of innate immunity mechanisms. Usually, whole BCG cells, not individual BCG cell fractions, are used in studies. Kovačić et al. (J. Med. Sci. 2021, 90(4):e551) citing Tanner et al. (Front. Immunol. 2019, 10:1317), Liu et al. (Human Vaccines, Taylor & Francis, 2009, 5(2):70-78), and Luca and Mihaescu (Maedica (Bucur), 2013, 8(1):53-58) state that there are insufficient data to show how BCG mycobacteria confer protective immunity and non-specific effects. The efficacy of the BCG vaccine may be related to differences within the individual BCG strains used for immunization. Some authors point to mannose-capped lipoarabinomannan (ManLAM), which promotes IL-8 production by macrophages, a cytokine responsible for neutrophil recruitment and activation (PLoS Pathog. 2018, 14(10):e1007358; Eur. J. Immunol. 2013, 43(8):2114-25; Scand. J. Immunol. 1992, 36(5):713-9; Nat. Immunol. 2014, 15(10):938-46; Cell. 2018, 172(1-2):176-190.e19). However, it should be noted that other molecular mechanisms participating in the response to BCG mycobacterial PAMPs may be involved in the development of immune mechanisms. BCG bacilli express different proteins that activate TLRs and activate macrophages and dendritic cells. Monocytes from BCG-immunized individuals have been shown to produce increased amounts of IL-1β, TNF and IL-6 in response to contact with unrelated pathogens, but no specific BCG cell component responsible for this has been identified, focusing instead on the induction of epigenetic and metabolic reprogramming of myeloid cells in vaccinated individuals (Nat. Rev. Immunol. 2020, 20, 335-337). Antigen 85, also present on the BCG surface, can induce the secretion of pro-inflammatory cytokines by the immune cells (J. Clin. Invest., 2018, 128:1837-1851; J. Immunol., 2008, 181:7948-7957).
3/ the information about symptoms of SARS-CoV-2 infection was removed from section 3.
We hope this improves readability of the paper.
Reviewer 3 Report
The review article entitled “BCG and SARS-CoV-2 – what have we learned?” by Kulesza et al compiles a very interesting observation and the various clinical trials that highlight the probable benefits BCG-vaccine may afford against SARS-CoV2 infections. The article is very well written and incorporates fair number of details on both BCG and SARS-CoV2, and the plausible mechanism why which the BCG vaccine may be exerting its protective effect. Section 4, while covering all the important points is dense and very lengthy. Perhaps it would be helpful to discuss some of the data in a tabular form making this section more succinct and maintaining the interest of the readers. It would also strengthen the article if the outcome of the trials mentioned in the conclusion is included, if available.
Author Response
The authors wish to thank the Reviewer for revising the manuscript and precious comments.
Regarding the studies referred to in the Conclusions, the estimated study completion date defined as a date on which the last participant in a clinical study was examined or received an intervention/treatment to collect final data range from December 2020 to May 2022. Unfortunately, the study results posted on ClinicalTrials.gov for the studies are not available.
We also added two subheadings in section 4:
4.1. Cross-protective immunity brings hope
4.2. An illusory hope…BCG does not change the COVID-19 status
We hope that the changes that have been made meet the reviewer's expectations and they improve readability of the paper.
Round 2
Reviewer 1 Report
Estimated Authors,
even though the present reviewer understand some of your points, I'm forced to stress that hardly one of my previous concerns had been addressed in this revised paper, including the critical issue represented by the potential claims of cherry picking for the content of this narrative review: the very nature of the study could at least be addressed in its conclusive remarks.
As a consequence, also my judgement cannot be changed, and I'm still recommending the rejection of this paper.
Author Response
“Cherry picking” is the leading issue raised in the review. According to encyclopedic definitions it is pointing to single instances or data that support a particular thesis or information, while ignoring a significant amount of other related ones that may be contradictory. The authors hope that, in the end, they will be able to convince the Reviewer that the assessed manuscript and its contents are as far as possible from the mentioned definition.
As mentioned in the previous reply, the evaluated paper cites both publications that point to a potential beneficial role of BCG vaccination in COVID-19 infection and publications that question such a relationship. To emphasize that, two subheadings have been introduced in section 4 that position the published research findings (4.1. Cross-protective immunity brings hope / 4.2. An illusory hope…BCG does not change the COVID-19 status). It was also repeatedly emphasized that we are dealing with suggestions and hypotheses. As required by the Reviewer, Chapter 3 has been shortened by removing some information (of a more historical nature) relating to the BCG vaccine, however further shortening of this chapter would have entailed a change in the conception of the work; information on "non-specific immunity" or "innate immune training" make the contexture of the prepared paper.
It is being increasingly claimed from scientific circles that the currently predominant strains of the pandemic virus differ so significantly from the baseline (2019) strain that it would be appropriate to start talking about a new disease entity, now sometimes referred to as COVID-22, instead of COVID-19, precisely to emphasize the antigenic differences that have occurred in the etiological agent that emerged in 2019. And yet, all the time, the agencies responsible for public health (WHO, CDC, ECDC, EMA) are recommending further vaccination with vaccines prepared on the basis of the initial variant of the virus. From an immunological point of view, the effectiveness of such action must be based on the phenomenon of cross-protective immunity. It is therefore not surprising that this phenomenon is included in both sections: 2 and 4. This does not at all imply that it was the intention of the authors to promote and/or prove the thesis that the BCG vaccine, which can also promote such a cross-protective phenomenon, can be a remedy for COVID-19. On the contrary, our article includes data obtained from numerous scientific papers aimed at investigating the potential impact of BCG vaccination on the COVID picture. Arguments “pro” and “con” have been presented. This was demonstrated by the inclusion of two subsections in Section 4. Moreover, the Abstract clearly states that, to date, there is no evidence that unquestionably and definitively demonstrates a beneficial effect of BCG vaccination in a SARS-CoV-2 pandemic situation. In the conclusion section, as requested by the reviewer, it is stated that epidemiological and observational studies have specific limitations and therefore can be a source of erroneous conclusions, hence hypotheses should be verified by randomized studies. The authors do not know how more clearly they should have expressed this, and they infer from the received review that this is the reviewer's expectation. The paper is not biased and the authors have not succumbed to the temptation of 'cherry picking'.
In order not to be accused of being even more biased, we refrained from including in the manuscript the results of the most recent study published in July 2022 in Frontiers in Immunology (Front. Immunol. 2022;13:873067.). This multicenter, double-blind trial (acronym ACTIVATE) included 301 participants aged 50 years or older receiving BCG vaccination or placebo. Covid-19 incidence and the presence of anti-SARS-CoV-2 antibodies within 6 months of BCG administration were assessed. Among other things, it was showed a 68% relative reduction of the risk to develop COVID-19, using clinical criteria or/and laboratory diagnosis, in the group of BCG vaccine recipients compared with placebo-vaccinated controls, during a 6-month follow-up (OR 0.32, 95% CI 0.13-0.79). The final conclusion was that BCG vaccination may be a promising approach against the COVID-19 pandemic (doi: 10.3389/fimmu.2022.873067). Furthermore, Rahali and Bahloul (Curr. Microbiol. 2022; 79(9):275) published also in July 2022 the research results indicating that BCG vaccination alone (mouse model) was able to induce cross-reacting antibodies to SARS-CoV-2 Spike. The authors concluded that both BCG vaccine immunization and latent tuberculosis infection may explain the lower burden of COVID-19 in developing countries, not only through the trained immunity but also by inducing cross-reacting antibodies (doi: 10.1007/s00284-022-02971-w). Moreover, the authors of the review publication in International Immunopharmacology, which appeared in the July 2022, explicitly encourage “more scientists to investigate further BCG induced non-specific immune responses and explore their mechanisms, which could be a potential tool for addressing the COVID-19 pandemic and COVID-19-like "Black Swan" events to reduce the impacts of infectious disease outbreaks on public health, politics, and economy”. This is supported by the following arguments: “1) the nonspecific protection effects of BCG, such as prophylactic protection effects of BCG on nonmycobacterial infections, immunotherapy effects of BCG vaccine, and enhancement effect of BCG vaccine on unrelated vaccines; 2) recent evidence of BCG's efficacy against SARS-COV-2 infection from ecological studies, analytical analyses, clinical trials, and animal studies; 3) three possible mechanisms of BCG vaccine and their effects on COVID-19 control including heterologous immunity, trained immunity, and anti-inflammatory effect” (doi: 10.1016/j.intimp.2022.108870).
With respect to the above-quoted examples, our article appears to be very conservative, with extremely cautious conclusions. Not only that, to erase all doubt of the Reviewer, subsection 4.2. was enriched with one more examples of research results that do not confirm the effectiveness of BCG in Covid-19 mitigation (reference no. 131).
We express our firm belief that the corrections made to the text of the manuscript and our clarifications will find understanding.
Reviewer 2 Report
The manuscript has appropriately been revised, and I have no serious criticisms.
Author Response
The authors once again thank you for undertaking an effort of reviewing our manuscript and for all valuable comments.